# New Discovery of Neogene Fossil Forests in Guatemala

**George E. Mustoe [1],\* and Markus Eberl [2]**

1   Geology Department, Western Washington University, Bellingham, WA 98225, USA
2   Department of Anhtropology, Vanderbilt University, Nashville, TN 37235, USA; markus.eberl@vanderbilt.edu
\*   Correspondence: mustoeg@wwu.edu

**Abstract:** Petrified wood specimens found at Maya archaeological sites are presumed to have been used during ceremonial fire-drilling. The source of this fossil wood has been an enigma, because in modern times no fossil wood localities were known to occur in Guatemala. In 2019, field work led to the discovery of two locations where silicified wood is abundant. Physical properties and microscopic characteristics of the fossil wood from the two sites have distinctive differences, suggesting that the geologic source of material used by the Classic Maya for cultural purposes can be recognized.

**Keywords:** Aguateca; archaeology; Central America; fossil wood; Guatemala; Maya

---

## 1. Introduction

This project began with an archaeological goal: to determine whether the petrified wood used in prehistoric Maya artifacts could be traced to specific geologic sources. Prior to our study, there were no published descriptions of petrified forests or sources of petrified wood in Guatemala, or the adjacent regions of Mexico, Honduras, and El Salvador.

In 1999, archaeologist Markus Eberl excavated petrified wood at the Classic Maya city of Aguateca in the tropical lowlands of Guatemala [1]. They come from a ritually used rock outcrop next to a cave-like chasm and date to 600–800 C.E. [2]. Seventy-four petrified wood fragments weighed about one kilogram. About a quarter of them could be reassembled single units that averaged ~15 cm in length and ~3 cm in thickness (Figure 1). Several tips were smoothly abraded; outer surfaces had blackened spots presumably from the exposure to fire. These observations suggest that the petrified wood sticks served as or imitated fire drills [3]. This use and their find context fit other evidence that the Classic Maya drilled fire in caves to venerate gods and ancestors. It remains unclear whether the Classic Maya used petrified wood similarly elsewhere. So far, petrified wood has been found only at two other archaeological sites [4,5].

Field work in 2019 resulted in the discovery of two areas in Guatemala where silicified wood occurs in abundance (Figures 2 and 3). Our preliminary studies suggest that the fossil woods from the two fossil forests can be distinguished based on physical properties and mineralogy.

### 1.1. Location Descriptions

The fossil wood localities are described as new discoveries because there has been been little public awareness of their existence. The few known instances of Guatemala petrified wood specimens being held in museums or private collections all involve materials that lack labelling for showing the collecting locality. The landowners of the localities described in this report were unaware of the occurrence of petrified wood on their properties.

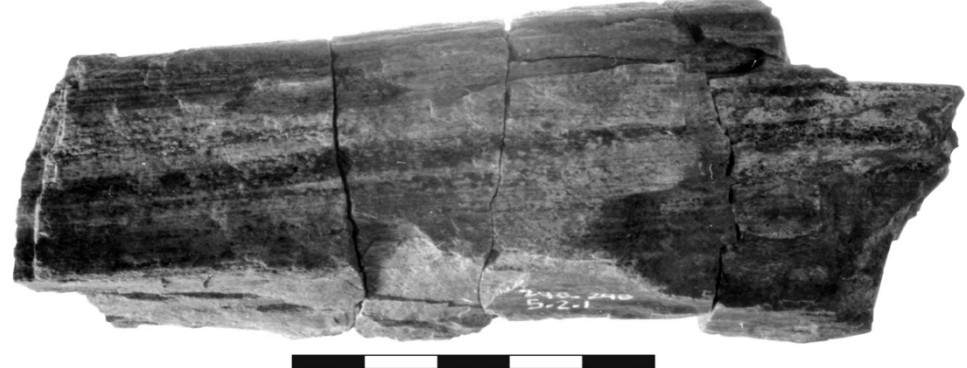

**Figure 1.** Petrified wood found during an excavation at Aguateca archeological site, showing the reassembly of angular fragments. Scale bar = 5cm.

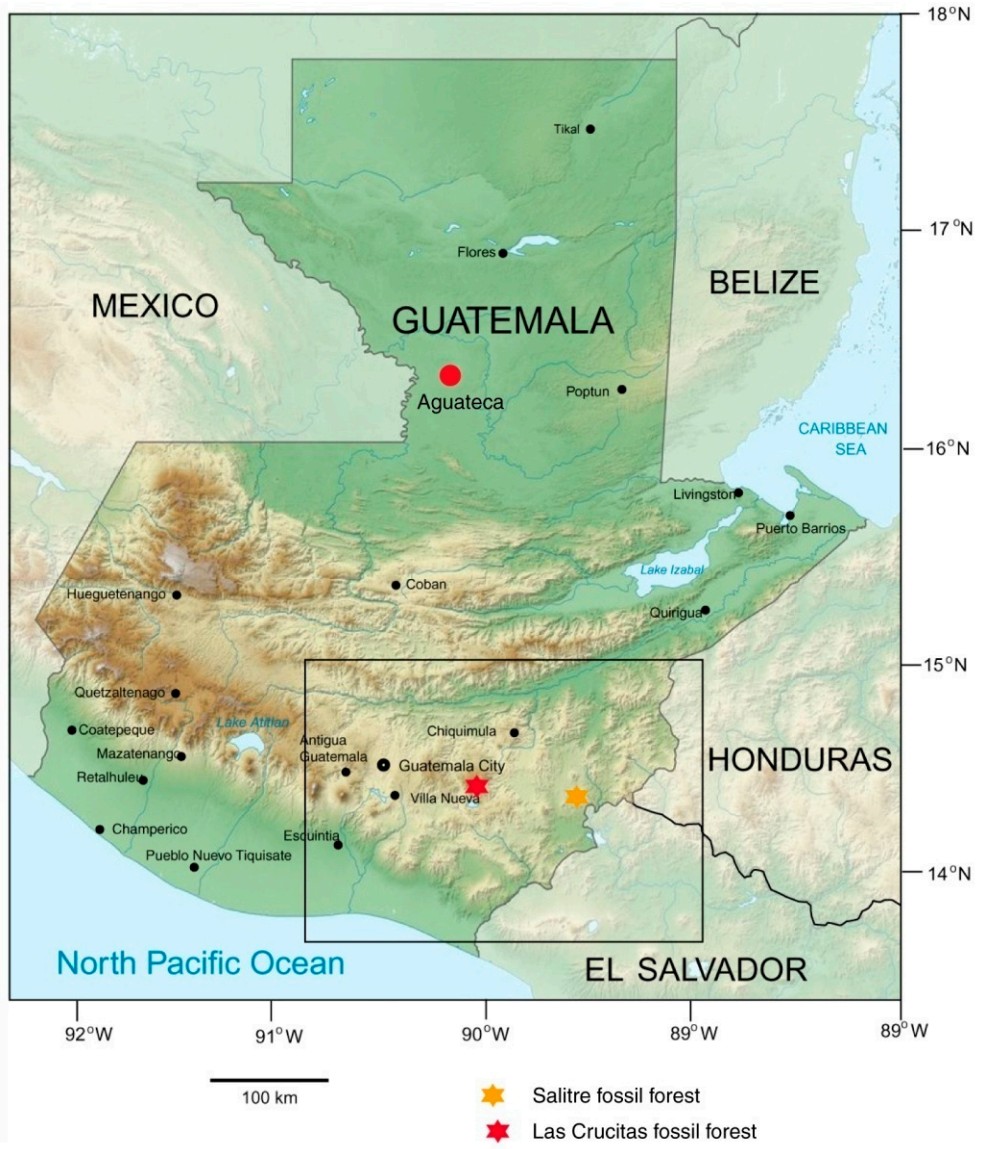

**Figure 2.** Location Map. The outlined area is the region shown in the map presented in the Geologic Setting Section.

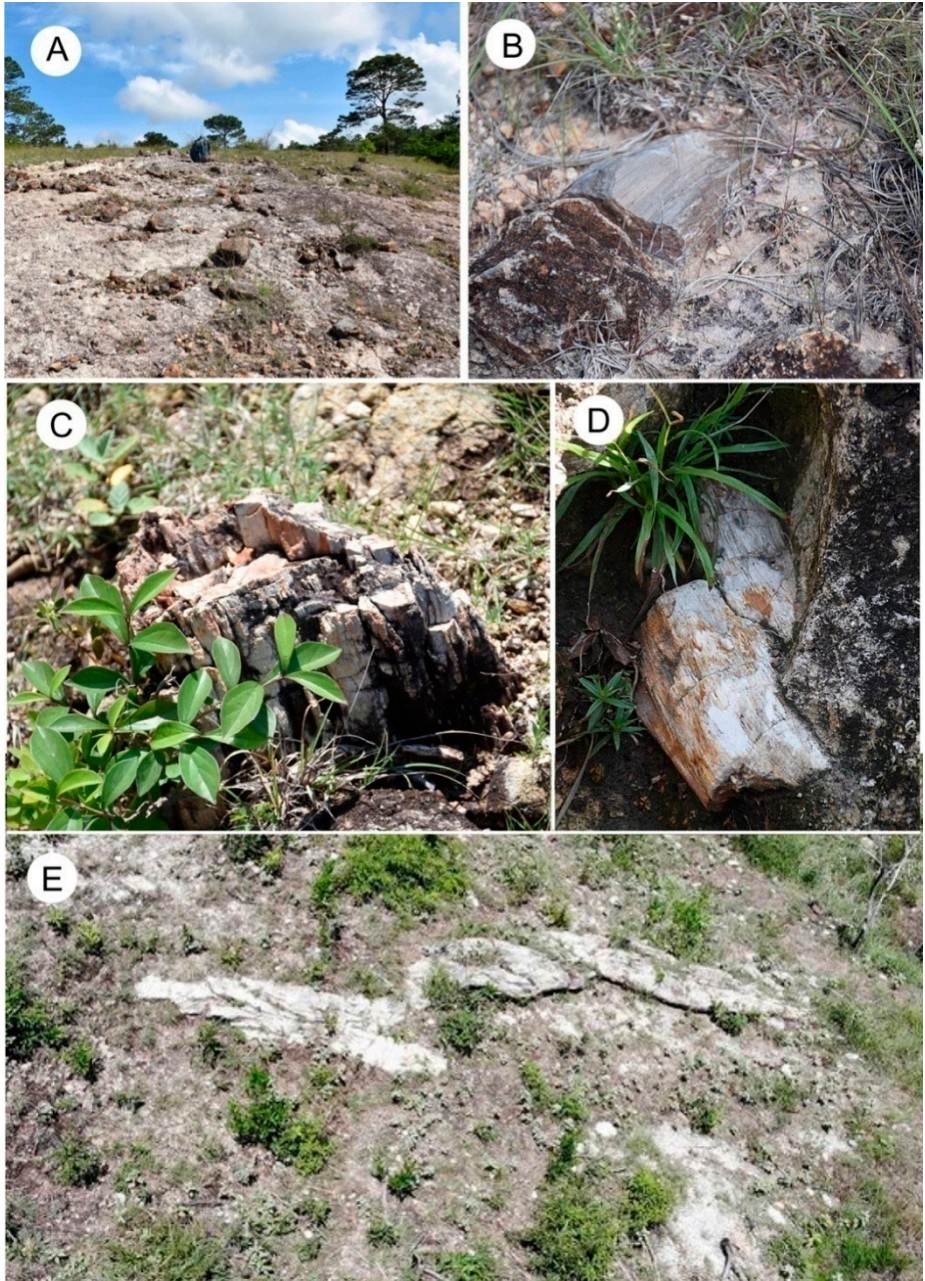

**Figure 3.** Field photos. (**A**) A rock outcrop at the Las Crucitas source with the highest concentration of petrified wood. (**B**) An example of a petrified wood fragment in place at Las Crucitas. (**C**) Petrified wood trunk at Salitre. (**D**) A petrified wood fragment enclosed by volcanic rock at Salitre. (**E**). A Y-shaped fossilized tree.

*1.2. Las Crucitas Locality*

A fossilized wood trunk has been exhibited in the city of Jalapa since 1963. A plaque specified the source area as the hamlet of Las Crucitas. Eberl located the last survivor who remembered where locals had found the trunk. Since the source area is now on privately own land, permission to enter the ranch was sought and granted. The landowner allowed surveying, collecting, and documenting but not the excavation of petrified wood fragments. Eberl and his team located dozens of petrified wood fragments on a hillside near the foot of the Jumay volcano and overlooking a lagoon (Figure 3). The surveyed area extends over approximately 20,000 square meters or 2 hectares and its elevation

ranges from 1399 m to 1450 m. The largest concentration of petrified wood fragments occurred on a rock outcrop; otherwise, grasses, bushes, and occasional trees cover the hillside almost completely. Bare volcanic rock from a hardened lava flow was observed two hundred meters away from the hillside. The largest petrified wood fragments are three trunks protruding out of the soil; all other fragments are finger-to fist-sized and dispersed loosely on the surface.

*1.3. Salitre Locality*

The second area is a petrified forest near the border between Guatemala and El Salvador.Salitre refers to the small river that divides the two countries. A teacher from the city of Agua Blanca guided Eberl and his team to the fossil site. The landowner gave permission to enter his property and guided the daylong survey. The area is located on a hillside in the vicinity of the volcanoes Ipala and Ixtepeque. Elevations range from 561 m to 689 m. The landowner uses the area for cattle-ranching because it is too rocky for other purposes. The surveyed area covers 30,000 square meters or three hectares; nonetheless, the landowner knows of more petrified wood fragments elsewhere. Dozens of petrified wood trunks were observed (Figure 3). In several areas, trunks are still vertically embedded in rock. Spaced at several meters apart, the trunks preserve the layout of the original forest. Most trunks measure 0.3 to 0.4 m in diameter. Exceptionally large were the Y-shaped remains of an ancient tree with a trunk diameter of 0.8 m and a length of approximately 10 m (Figure 3E). Innumerable petrified fragments cover the ground.

Although large specimens are preserved at both locations, specimens collected for this study are small, selected as being representative of the range of materials found at each site (Figure 4).

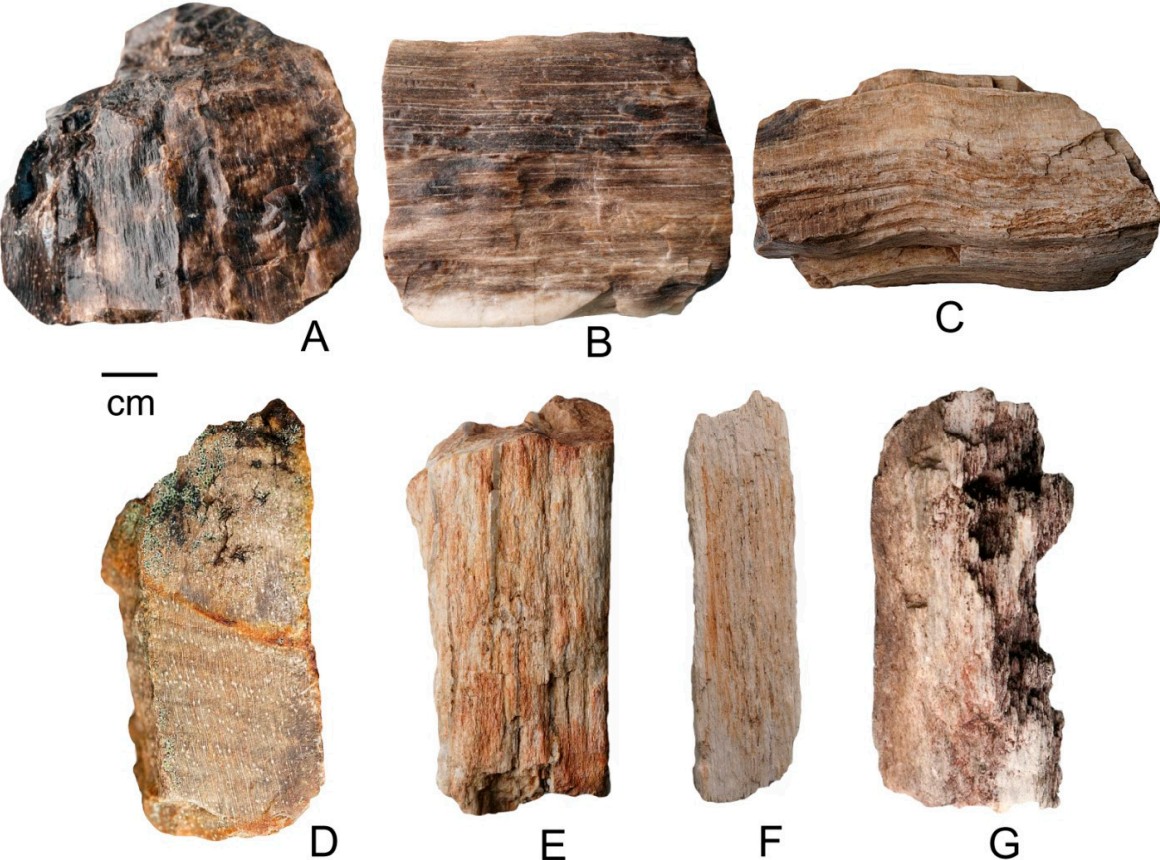

**Figure 4.** Examples of fossil wood specimens used in this study. (**A–D**) Las Crucitas locality; (**E–G**) Salitre locality. All specimens are tropical gymnosperms, except for (**D**), which is an angiosperm.

## 2. Geologic Setting

The tectonic history of Central America involves the complex interactions of several crustal plates (Figure 5). In Guatemala, the northern part consists of igneous and metamorphic basement rocks of Paleozoic age, which are overlain by Mesozoic sedimentary rocks. Tectonic compression has caused these beds to have been folded, faulted and uplifted to form the Central Guatemala Cordillera. Younger deposits include early Tertiary volcanic and marine clastic rocks. In southern Guatemala, the bedrock is part of the ChortisBlock, a detached segment of the Caribbean Plate that was translated eastward during the late Mesozoic. The Cocos Plate became separated from the main Pacific Plate in the Oligocene, resulting in a subduction zone that produces the "ring of fire" volcanoes that parallel the Pacific Coast. In Guatemala, this sequence presently includes 37 volcanoes that have been active in the Holocene, including several that have undergone explosive eruptions in modern times [6–8].

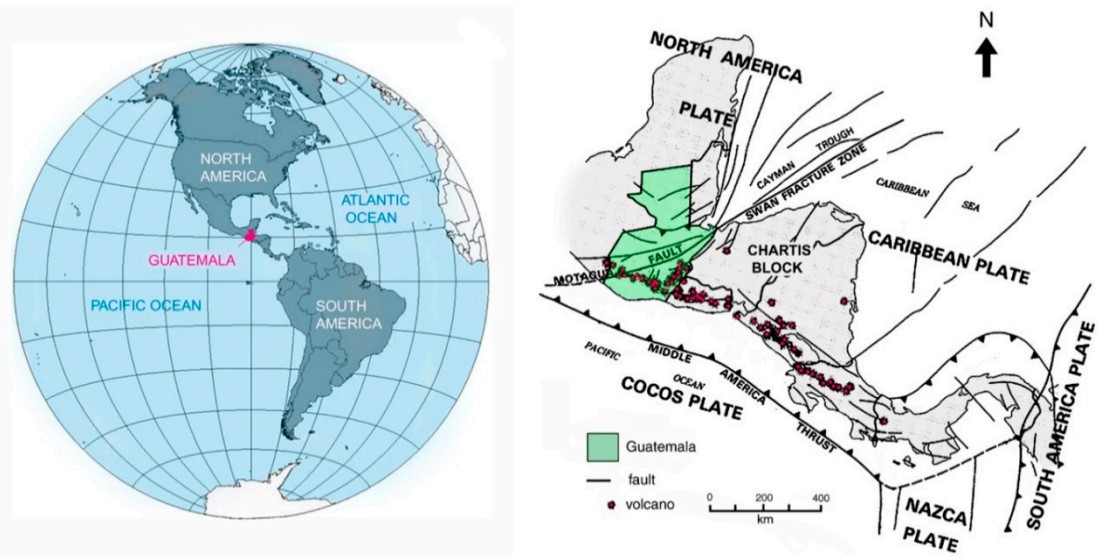

**Figure 5.** Generalized tectonic architecture of Central America. Modified from [6].

Unlike other Central American countries, Guatemala and its neighbor Belize contain extensive sedimentary rocks. In Guatemala, these sediments comprise about 60% of the country's bedrock, mostly in the northern half, extending into the Yucatan Peninsula of Mexico [6]. These sedimentary rocks primarily consist of marine deposits, including redbeds, evaporates, and thick limestone layers. These strata locally preserve invertebrate fossils, but known occurrences of fossil wood are limited to the volcanic zone, where the explosive eruption of felsic magma produced extensive pyroclastic flows that provided favorable conditions for the rapid burial of tree trunks, and provided a source of dissolved silica for subsequent fossilization.The stratigraphy of these volcanic deposits has not been mapped in detail, but the age of the wood-bearing strata is presumed to be late Tertiary. The general geology is shown in Figure 6.

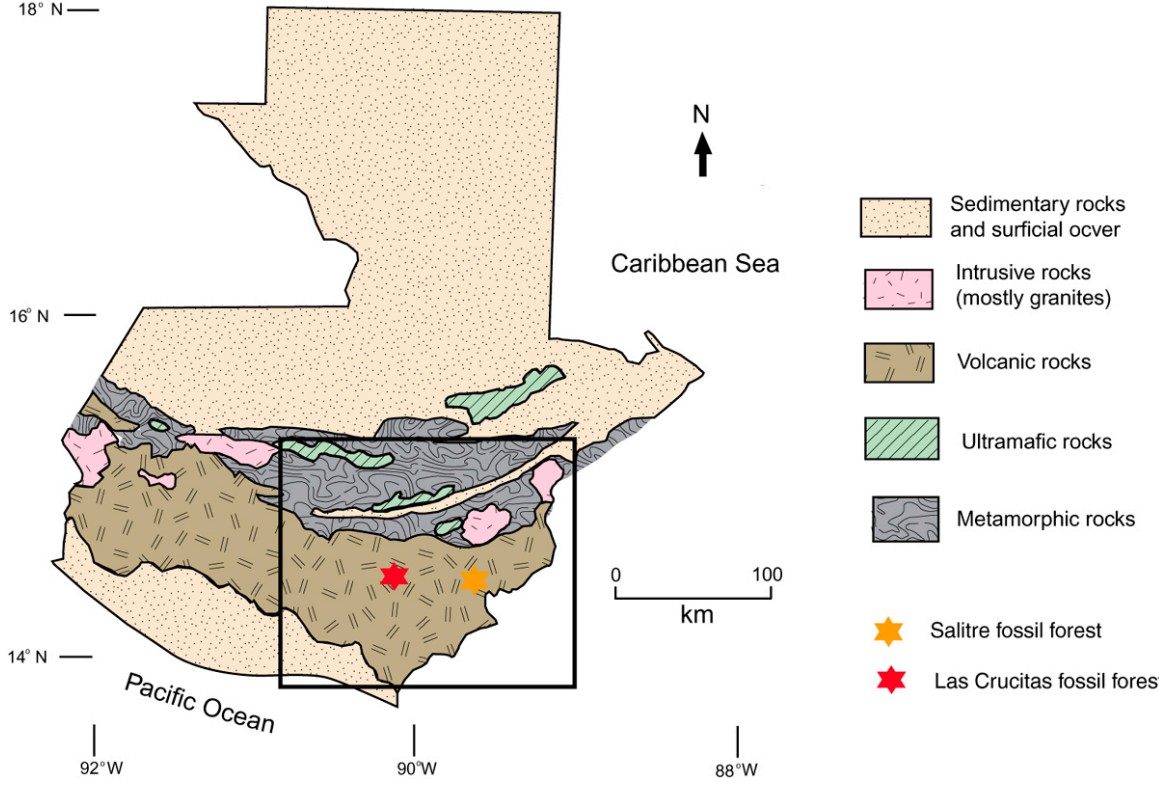

**Figure 6.** Geologic map of southeast Guatemala. Modified from [9].

## 3. Methods

This report describes physical and optical characteristic on 5 petrified wood samples from the Salitre site and 5 samples from the Las Crucitas locality. Scanning electron microscope images were made using a Tescan Vega scanning electron microscope. Optical photomicrographs were obtained using a Zeiss petrographic microscope equipped with a 5 megapixel digital camera. Density values were determined using a hydrostatic weighing device installed on a Mettler analytical balance, as described by [10]. Approximately 15 specimens were collected from each site. Specimens and location information are archived at the Anthropology Department, Vanderbilt University.

The unusual challenges of geologic field work in Guatemala deserve mention. In the past, silver, nickel, and uranium mining has caused environmental damage. Local residents have been protesting against these mining efforts because the government commonly offers mining concessions without consulting the communities affected by the development. These conflicts have at times escalated to the point where fatalities occur. Locals may become defensive if outsiders inquire about geologic conditions. Another concern is the risk that fossil sites may be looted if discoveries are publicly disclosed. In Guatemala, governmental protection of natural historic resources is limited.

## 4. Results

### 4.1. Salitre Fossil Wood

Petrified wood from the Salitre locality is relatively soft, and cuts readily with a diamond lapidary saw. Samples are light-colored, typically whitish with light red iron oxide stains. End grain surfaces absorb water, indicating porosity. The light color suggests that very little relict organic matter is present. The luster of freshlybroken surfaces is rather dull, resembling unglazed porcelain.

In SEM images, Salitre wood shows cell walls that are mineralized with microcrystalline quartz (Figures 7 and 8). Cell lumen are mostly quartz-filled, but some remain empty (Figure 9). Intercellular

spaces commonly remain unmineralized, which explains the absorption of water on end-grain surfaces. Thin sections viewed by transmitted light optical microscopy (Figure 9) commonly show dark-colored cell walls that are isotropic (i.e., opaque black) in polarized light views. The most likely explanation is that the relict organic matter produces dark color in ordinary transmitted light views, but the incipient silicification of these tissues causes the isotropic opacity in polarized light images. The cell interiors (lumina) are filled with microcrystalline quartz. Because pure quartz has a density of 2.60 g/cm$^3$, the lower densities measured for Salitre wood specimens are probably caused by the empty intercellular spaces (Table 1).

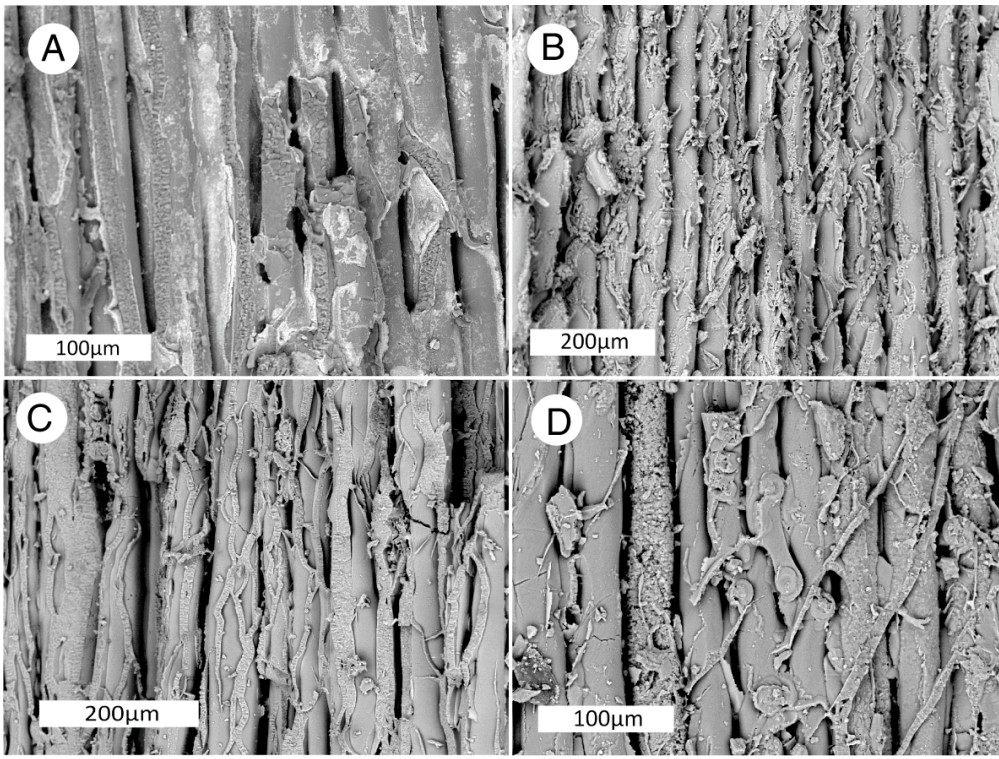

**Figure 7.** Scanning electron microscope images of Salitre petrified wood, showingunmineralized paces between cells. (**A**) Specimen S1A. (**B**) Specimen S1-C. (**C**) Specimen S2-A. (**D**) Specimen S2-B.

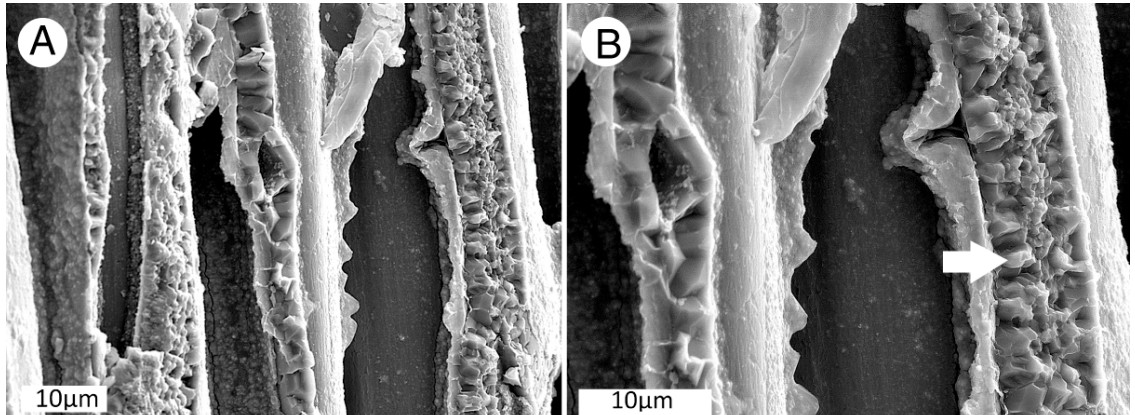

**Figure 8.** SEM images of Salitre specimen S1-A showing empty lumina. (**A**) Three adjacent tracheids with empty lumina; (**B**) The cell walls have been mineralized with microcrystalline quartz (arrow).

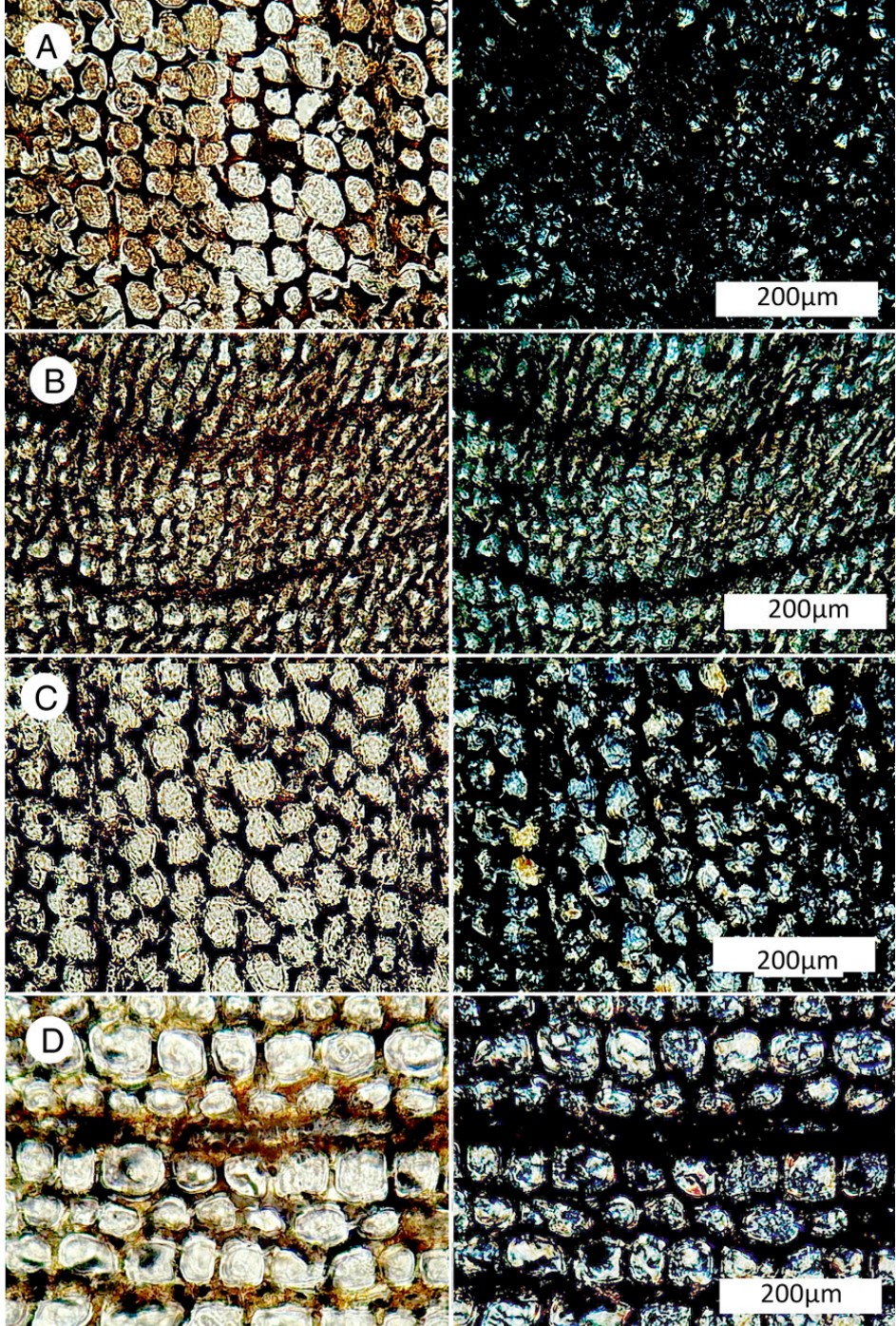

**Figure 9.** Optical photomicrographs of thin sections of transverse views of Salitre fossil wood. The thickness is approx. 30 microns. For each photo pair, the image on left is ordinary transmitted light, and the photo on right is the polarized light view. The cell walls are dark colored in ordinary light images, and opaque (isotropic) in polarized light images. The cell lumen are filled with microcrystalline quartz. All images are taken at the same magnification. (**A**) Specimen S1-A. (**B**) Specimen S1-B. (**C**) Specimen S1-C. (**D**) Specimen S2-A). The absence of conductive vessels suggests these Salitre specimens are conifers. The lack of prominent annual rings suggests a paleoclimate that was equable, i.e., not strongly seasonal.

**Table 1.** Density of fossil wood.

| Sample | Density g/cm$^3$ | Comments |
|---|---|---|
| Salitre: | | |
| S1-A | 2.20 | Slightly porous |
| S1-B | 2.47 | Slightly porous |
| S1-C | 2.12 | Slightly porous |
| S2-A | 2.10 | Slightly porous |
| S2-B | 2.06 | Slightly porous |
| Las Crucitas: | | |
| LC-A | 2.49 | Non-porous |
| LC-B | 2.35 | Non-porous Contains opal and quartz |
| LC-C | 2.54 | Non-porous |
| LC-D | 2.47 | Non-porous |
| LC-E | 2.45 | Slightly porous |

*4.2. Las Crucitas Fossil Wood*

Las Crucitas specimens are harder than the Salitre samples, cutting at a slower rate with a diamond saw blade. The luster of freshly-broken surfaces is rather dull. Specimen colors range from whitish to medium brown, the darker specimens typically containing lighter areas. Wood grain textures are evident on sawn surfaces.

Transmitted light optical microscope views of petrographic thin sections (Figure 10) reveal that most samples consist of nearly pure quartz. In some specimens, cell walls are dark when viewed under transmitted light, and isotropic under polarized light. In other specimens, cell walls are nearly colorless, indicating that the original organic matter has been replaced by quartz. Densities of Las Crucitas fossil woods are notably higher than for Salitre specimens (Table 1). Low densities of Salitre specimens can be attributed to their porosity. Las Crucitas specimens are typically non-porous, and variations in density are mostly related to variations in mineral composition.

The SEM images (Figure 11) show that the Las Crucitas fossil wood is typically fully mineralized, without the empty intracellular spaces that are so common in Salitre wood (Figure 8).

Specimen LC-B is an anomaly because it contains extensive regions that are isotropic in polarized light views, which is evidence that the wood is partially mineralized with amorphous opal (opal-A). Quartz mineralization is also present. Even in a single 1 cm$^2$ area of a thin section, the composition ranges from mostly opal to mostly quartz (Figure 12). The relatively low density of this specimen (2.35g/cm$^3$) suggests that opal is a major constituent. The origin of this heterogeneous mineralogy is enigmatic. The diagenetic transformation of amorphous opal to chalcedony and quartz is well-documented for silica in hot spring sinter and biogenic sediment (diatomite and radiolarian ooze). For silicified wood, the situation is more complicated: multiple silica phases can result either from solid-state transformation, or from multiple episodes of mineral precipitation. In the case of Las Crucitas specimen LC-B, opal-filled cells are often adjacent to quartz-filled cells, a characteristic that is difficult to explain as a product of diagenetic alteration. The physical and chemical gradients that facilitate the transformation of opal to quartz are not likely to occur on a cell by cell level. However, sequential mineralization involving discrete precipitation episodes is also difficult to envision as a cause of the differing mineralization of individual cells. One possibility is that the differing mineral compositions are related to the partial drying of the wood prior to fossilization, where some tracheid lumina became air-blocked, preventing the flow of mineral-bearing groundwater. Increased water saturation may have been achieved later by the partial breakdown of cell walls, allowing the entry of silica-bearing fluids

under geochemical conditions that were different from the initial silicificationepisode. This explanation is very speculative. The indisputable fact is that for whatever reason, sample LC-B has an anomalous mineral composition compared to other Las Crucitas fossil woods.

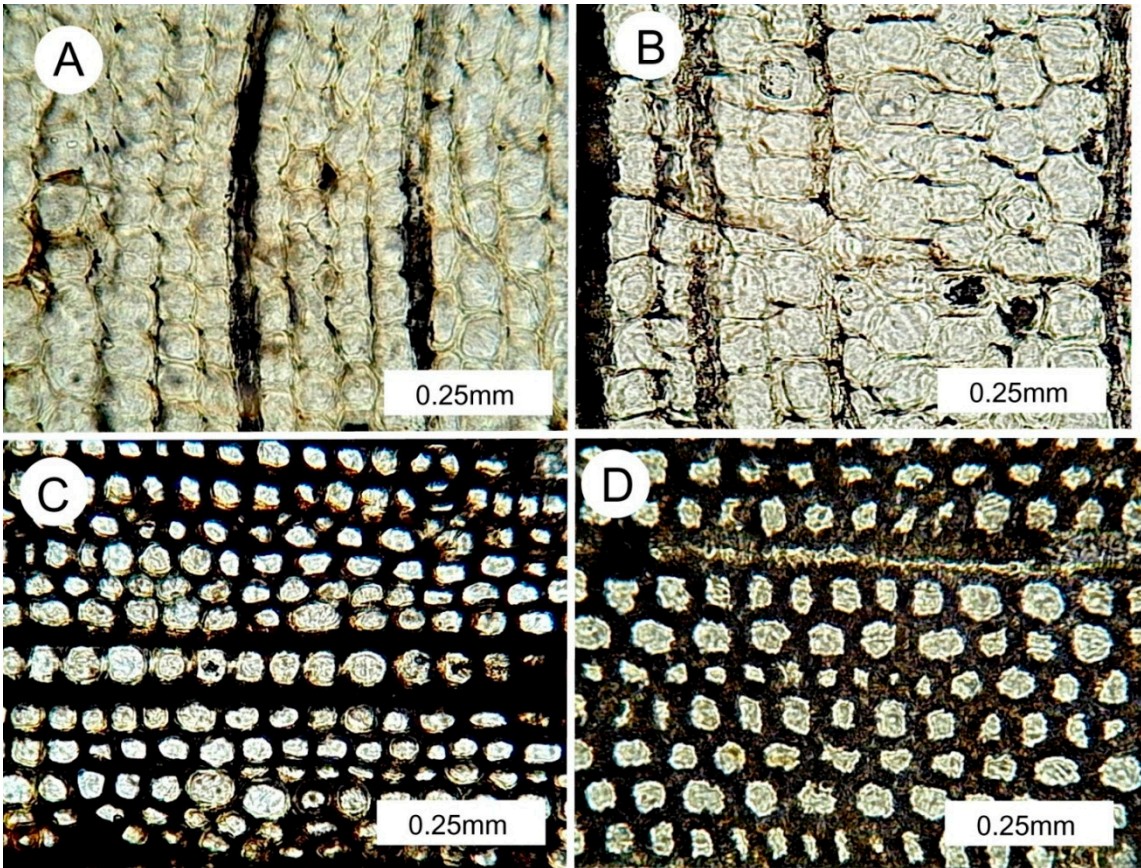

**Figure 10.** Transmitted light optical photomicrographs of thin sections of transverse views of Las Crucitas fossil wood. The thickness is approximately 30 microns. The absence of vessels is an indication the woods are all from gymnosperms. (**A**) Specimen LC- B. (**B**) Specimen LC –C. (**C**) Specimen LC-E. (**D**) Specimen LC-B. Cell lumina are filled with quartz. In specimens A & B, cell walls are mineralized with transparent quartz. Specimens LC-C and LC- D have dark colored cell walls that appear to consist of a combination of amorphous opal and relict organic matter.

The anatomical preservation of Guatemalan fossil woods is highly variable. As noted earlier, specimens from the Salitre locality typically have relatively poor preservation. As a general rule, the quality of anatomical preservation is related to the rate of mineralization relative to the rate of tissue degradation. Under conditions of rapid wood decay, cellular details may be destroyed before silica can replicate anatomical features. The absence of vessels in transverse views indicate that these fossil woods represent gymnosperms, and the lack of distinct early wood/ late wood "rings" are evidence of an equable climate. Otherwise, taxonomic interpretation is difficult. Fossil woods from the Las Crucitas locality typically have reasonably good preservation of cellular anatomy (Figure 13). These samples illustrated in this report are gymnosperms, but several dicot woods specimens were collected. The taxonomic identification of Guatemala fossil wood is a goal of research that is currently being organized for the 2020 field season.

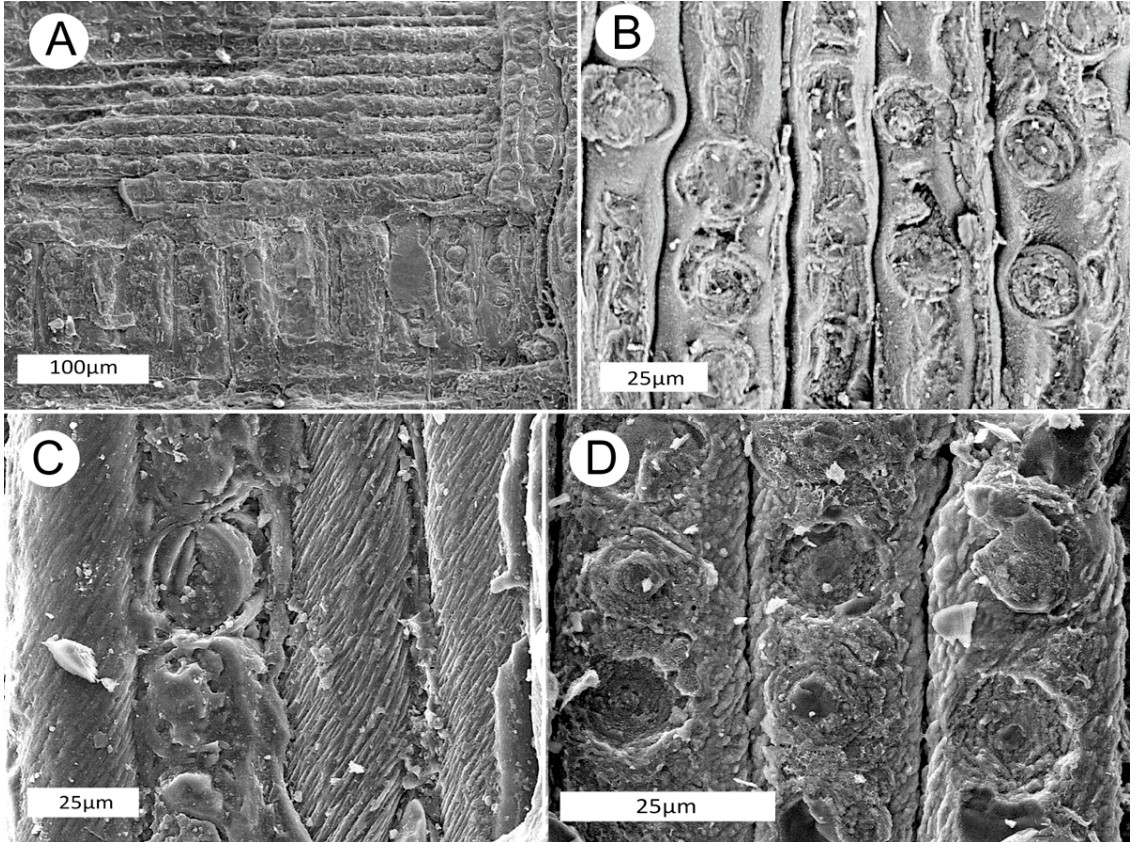

**Figure 11.** SEM images of Las Crucitas fossil wood. (**A**) Specimen LC-C, radial view showing horizontal rays. (**B**) Specimen LC-E, radial, showing uniseriate pitting. (**C**) Specimen LC-B, tangential view. Circular features are cross-section views of ray cells. The spiral textures on adjacent cells are evidence of microbial degradation. (**D**) Specimen LC-E, radial view showing pit casts. These specimens all show homoxylous architecture (i.e., an absence of vessels and fibers), evidence that the woods are from gymnosperms.

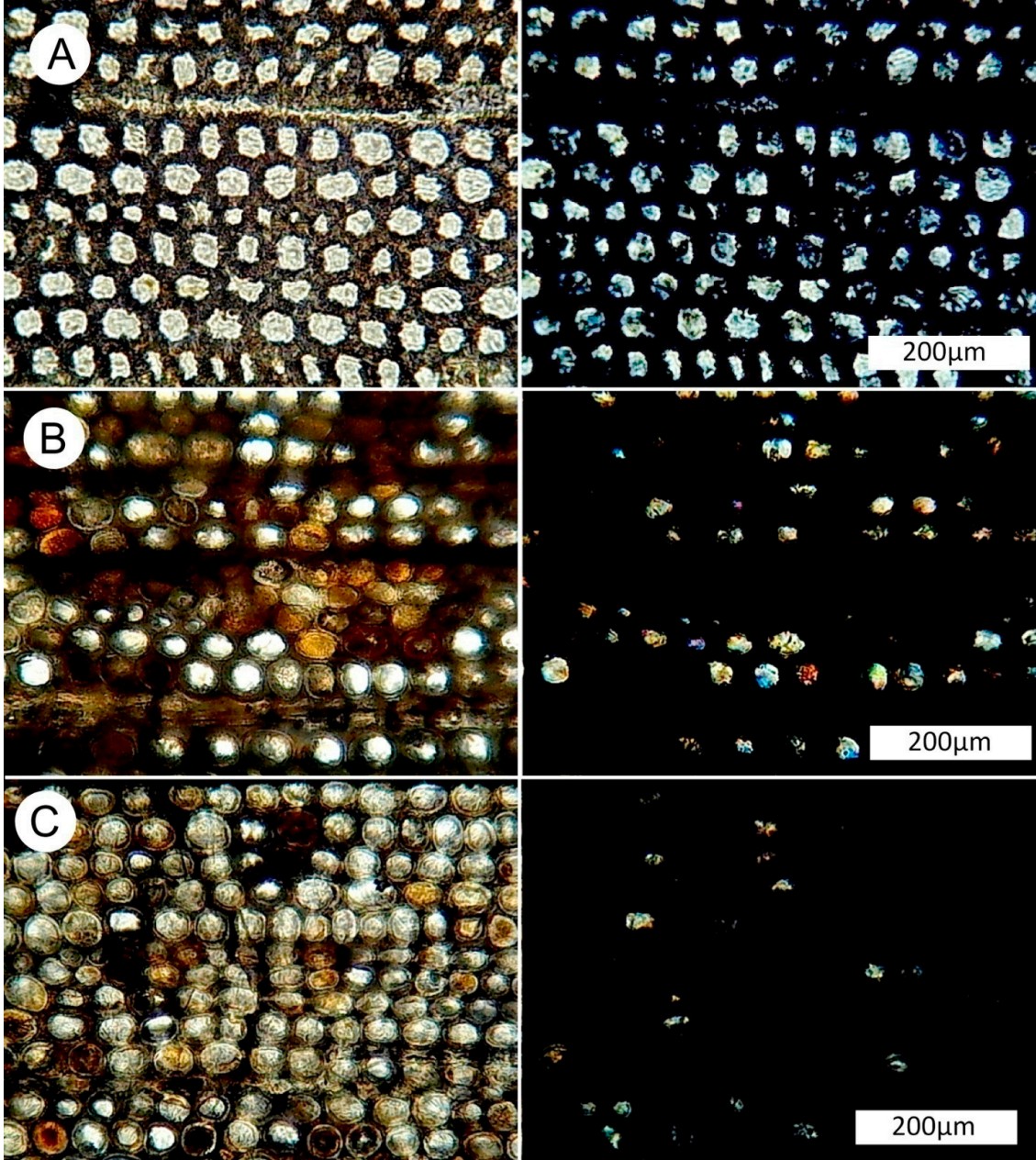

**Figure 12.** Optical photomicrographs of Las Crucitas sample LC-B. For each photo pair, the image on the left is ordinary transmitted light, and the photo on right is a polarized light view. The opaque black (isotropic) areas in the polarized light images are presumed to consist of amorphous opal (opal-A). The brighter areas are quartz that fills some cell lumina. Transverse views. (**A**) All cell lumina are quartz-filled. (**B,C**) The cells that show brown filling material in ordinary transmitted light are opal-filled, as evidence by their character under polarized illumination. Cells that are colorless in ordinary light are revealed by their interference colors under polarized light.

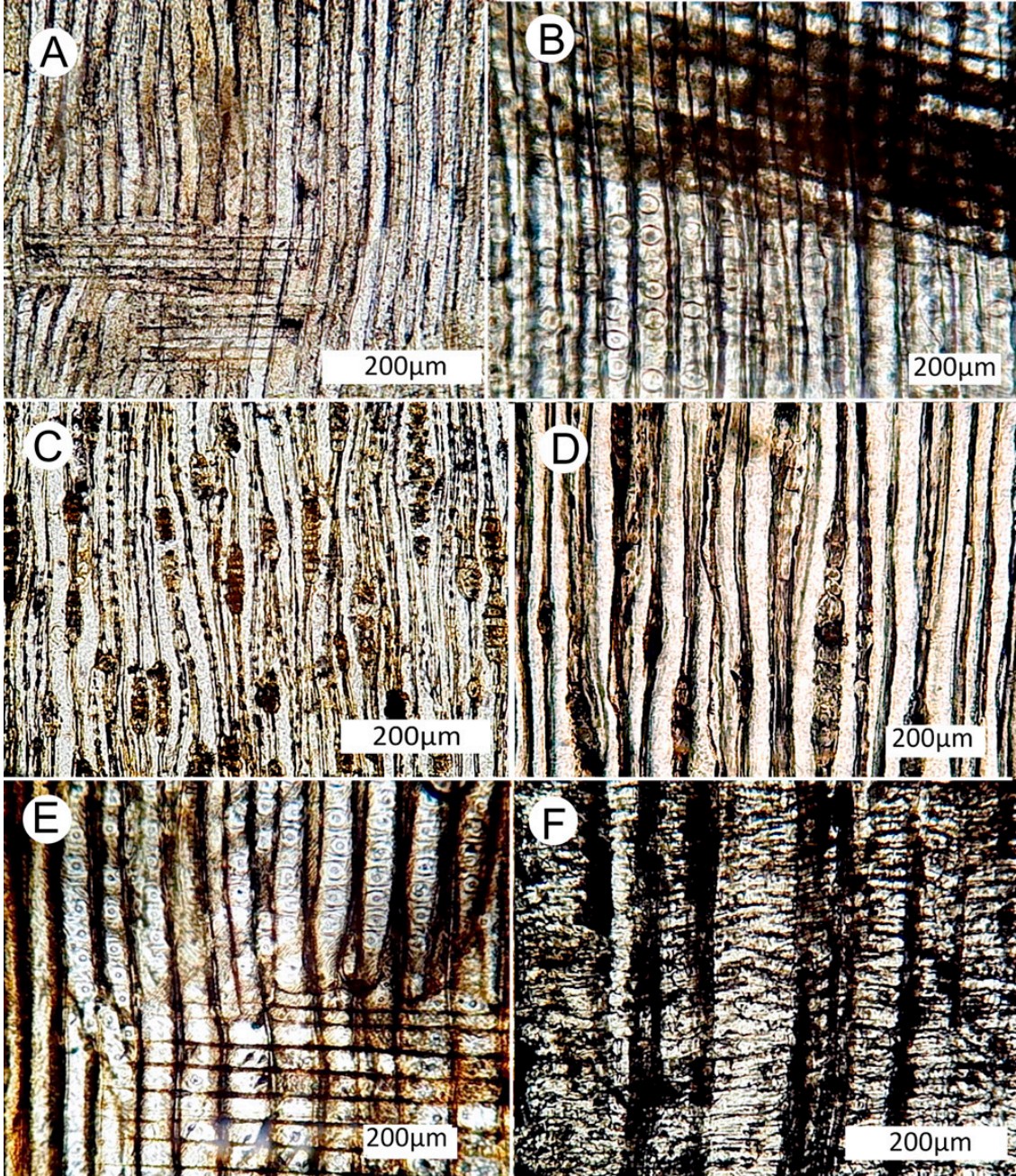

**Figure 13.** Cell anatomy. All specimens are from gymnosperms. (**A**) Specimen LC-B, radial view showing horizontal ray cells. (**B**) Specimen LC-B-radial view showing bordered pits typical of gymnosperms. (**C**) Specimen LC-C, tangential orientation revealing cross-section views of ray cells. (**D**) Specimen LC-D, tangential view showing ray cell cross-sections. (**E**) Specimen LC-E, radial view displaying horizontal ray cells and vertical tracheid cells containing uniseriate bordered pits. (**F**) Specimen LC-C, radial view showing ray cells as vertical lines and horizontal tracheids.

## 5. Discussion

The original goal of this study was to evaluate whether it is possible to recognize the source locality of petrified wood used for artifacts by Guatemala's pre-Columbian Maya. This study is based on only five small specimens from each of the two localities. However, these samples are adequate to support general interpretations; the full range of compositional variation cannot be ascertained without more detailed research. Our preliminary evidence suggests that specimens of fossil wood from the two

localities can be discriminated with reasonable confidence based on optical and electronmicroscopy, and by the resistance of the specimens to sawing. However, these are all destructive analytical methods that are not suitable for archaeological materials. Fortunately, the measurement of density is a useful indicator of fossil wood sources, and this is a nondestructive method. The measurement of density simply involves measuring the mass of the specimen in air, and suspended in water. In general, the densities of Salitre specimens are less than 2.20 g/cm$^3$ and greater than 2.35 g/cm$^3$ for Las Crucitas specimens. In addition, porosity can be evaluated during density determinations because samples that have permeability will show a gradual increase in measured weight when the fossil wood is suspended in water. As noted in Table 1, Salitre fossil woods are typically slightly porous, and lower in density compared to the non-porous woods from Las Crucitas. The density method is not perfect, because both groups contain a single outlier. Based on density alone, the Salitre sample S1-B could be mistaken for a Las Crucitas sample, but the light color and slight end grain porosity are typical ofSalitrespecimens.The Las Crucitas samples LC-B is unusual because of the abundance of opal and the anomalously low 2.45g/cm$^3$ density. However, the hardness, and impermeability of LC-B are similar to other Las Crucitas specimens. The sample LC-E is unusual because of its slight porosity, but the 2.45g/cm$^3$ density is typical of the Las Crucitas assemblage.

The differing physical properties of specimens from the two sites may have been a determining factor for the choice of petrified woods that were used for used for manufacturing artifacts. The relative softness of the Salitre wood would make the material potentially suitable for artistic carvings, but this fossil wood would not be suitable for applications that required a high polish, or the ability to hold a sharp edge. In contrast, the Las Crucitas wood would be more difficult to shape because of its greater hardness, but this hardness would be an asset for objects that required greater durability, or objects that were polished to produce a glossy surface. The Salitre locality is in a region that is known to have been a source of obsidian for ancient Maya [11], but no evidence of ancient extractive activities was observed at either fossil forest during out preliminary studies. We have not inferred a source for petrified wood previously found at archaeological sites, because continuing field work may reveal additional fossil wood localities. For the time being, our strategy is to establish possible methods for recognizing the geologic source for artifacts, not to apply our preliminary results to Maya artifacts.

*Possible Future Research*

Although the original goal of our research was archaeological, the abundance of fossil wood at the two localities suggests that these fossil sites may be important for other reasons. Petrified wood localities have not previously been reported from Guatemala. Indeed, with the exception of Panama, fossil wood has rarely been described from Central America. Silicified wood occurs in Oligocene and Miocene at several locations in Panama. This material has been described in detail by several research teams [12–18]. Panama fossil wood has been recognized as an important indicator for understanding the interchange of plant taxa between North and South America, and for interpreting the effects of tectonic change on plant communities. For example, the occurrence of plant fossils in the Miocene of Panama suggests that the intercontinental exchange of tree species occurred at least 10 million years before the date that has usually been accepted for the final closure of the Panama isthmus [13]. Additionally, the Panama fossil woods have proven important for the study of the paleoecology of neotropical rainforests. Finally, these fossils provide insights for understanding the relationships between extant plants and their Cenozoic ancestors.

The newlydiscovered fossil forests of Guatemala are potentially of great scientific interest for the same reasons as that apply to Panama fossil wood. The geographic location is important for evaluating the migration of plant taxa through time, because Guatemala would have been a gateway between North America and Central America. The translocation of plants from southern North America to Central America and South America would presumably show the arrival of these taxa in Guatemala earlier than in Panama. For the migration of plants from south to north, the arrival times would be reversed. Several factors will be important for future investigations. Detailed geologic mapping,

stratigraphic correlation, and radiometric dating are needed to provide evidence for determining the age of petrified wood localities. Careful taxonomic identifications are needed; well-preserved cell features in Las Crucitas samples suggests that taxonomy can be determined for at least some specimens from this locality.The lower quality of preservation of the Salitre specimens may limit their potential for detailed identification.

Although our preliminary research has focused in archaeological aspects, the study of fossil forests in Guatemala has other implications. Climate change is disrupting agriculture, causing many residents toattempt to flee to the United States because they can nolonger support themselves and their families [19]. Both fossil forests are located in Central America's dry corridor that is becoming increasingly more arid. The evidence from Neogene tropical forestcan help Guatemalans to better understand how dramatically their environment has changed.

**Author Contributions:** Field work, M.E.; Laboratory studies, G.E.M.; Writing, G.E.M., M.E. All authors have read and agreed to the published version of the manuscript.

**Funding:** Field work was funded by National Geographic Society Grant NGS-5345-19, "Harnessing Fire Mountains, Sourcing Petrified Wood in the Maya area". Opinions expressed in this article are those of the authors, and are notintended to represent the National Geographic Society.

**Acknowledgments:** We thank NareeratBoonchai and Marc Philippe for their paleobotanical insights regarding the possibilities for taxonomic analysis of Guatemala fossil wood, and for help in planning future research.

**Conflicts of Interest:** The authors declare no conflicts of interest.

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
