# Peer review of "New Discovery of Neogene Fossil Forests in Guatemala"

_geosciences, doi:10.3390/geosciences10020049_

Round 1

Reviewer 1 Report

This is an interesting article bringing attention to the occurrence of petrified woods among archaeological artifacts and providing an approach to try to determine their source area.

By Late Tertiary in the title, do you mean Neogene?  The latter term is favored by stratigraphers.

"three trunks still growing out of the soil"  This phrasing could be misconstrued as indicating the fossil stumps are living.   Replace "growing" with "protruding" or reorganize to say stumps are upright, apparently in situ.

Caption to fig 3.  Delete "Salitre. Access to both sites was by permission of the property owners."  This is irrelevant to the science and repeats what is already stated in the main text.

Figs. 9b, 9d, 10c, 10d, 12a-c.  These images of wood cross sections are in nonstandard orientation, with rays horizontal instead of vertical.  This could be corrected (if desired), by rotating the indicated images 90 degrees.  Same for 13f.

Caption to fig. 13.  "All specimens are from diffuse-porous gymnosperms..."  Delete "from diffuse-porous".  This is a term for angiosperms (pores are vessels, which are lacking in conifers).  It is usual for gymnosperms to have tracheids of uniform size through each growth ring.

line 249.   Unnecessary font change.

Following on the questions raised in the introduction, it would be nice to know if the authors can hypothesize which of these two sites is the more likely source of the archaeological samples.  I did not come away with a sense whether it is considered likely that one or both of these sites is the actual source of the woods from the archaeological site.  Do the authors consider it likely that other similar wood sources would be found with additional field work?  Some discussion about this in the concluding part of the manuscript would be welcome.  Although specific gravity is mentioned as a potential distinguishing method, this information is not provided for the archeological samples in question.

Author Response

Thanks for the helpful and encouraging review. We have paid careful attention to you comments and suggestions for improving the manuscript. The manuscript has been modified to accommodate all of your recommendations. The only exception is the non-standard orientation of the images that show tangential orientations, where the ray cells are horizontal rather than vertical. Because our story is being presented from a geological rather than botanical perspective, and submitted to a general readership journal we have opted to leave the illustrations in their present format. The present format was chosen to to orient photomicrographs so that tracheids are vertical, because that is the orientation of these cells in living plant stems. For tangential views, orienting the images so ray cells are vertical means that the tracheids are horizontal. I'd prefer to have the same structural orientation for all of the fossil wood images. This might be problematic for a paper written for botanists, but my belief is that the present photo format is the most understandable one for general  readers.

Many of the comments involve rather minor editing and wording issues. We appreciate these suggestions, and have made all of the suggested changes. More general comments refer to questions about how our work applies to actual archaeology sites. Some explanations have been added to the discussion section to provide more clarity for this issue.

Thanks again,

George Mustoe

Reviewer 2 Report

Mustoe and Eberl provide an initial study of two assemblages of permineralized woods from Guatemala. To my knowledge, this is the first study of permineralized woods from Guatemala and is thus a significant contribution to the literature. Future work on these fossils and their geological context will have important implications for archaeology, paleontology, and geology of Central America. As a reader, I appreciate that the inspiration for the study was the discovery of permineralized wood associated with an archaeological site. I would suggest (but certainly not insist) that the authors consider an alternative framing because the results of this investigation do not seem to resolve the question of where the artifacts were from. Instead, it simply shows that preservation of fossil woods vary from site to site and therefore we may be able to match the artifacts to a site in the future. These seems like excellent content for a discussion. The big picture for this study is that fossil woods have not been reported from Northern Central America, and they have the potential to provide valuable information about paleoenvironments and taphonomy. I would have liked to see more work on the identification of the woods, an attempt to determine their age, or a more detailed description of the type of rock in which some of the specimens are embedded to compliment the discussion of the mineralization process; nonetheless, I think it is worthy of publication as an initial study once some minor issues are addressed.

The manuscript is well-written overall, but I have made a handful of suggestions that might improve clarity. There are a few claims that should be supported by citations, and I have indicated these throughout. The methods and geologic setting are adequate, but some additional information is necessary. Where will the collections ultimately reside in the US or Guatemala? Will detailed locality information be stored there? How many samples were collected at each locality?

Some of the woods at Las Crucitas are described as “still growing out of the soil” (line 62), but it is unclear what this means. Are these trunk bases with roots that are still entombed in a paleosol horizon in original growth position? Are they dispersed wood fragments that are weathering out of rock matrix but not in growth position? Or are these permineralized woods weathering out of a modern soil? Lines 269-270 seem to indicate that the authors think they have found the sites where the Maya obtained the wood that was found in the archaeological site, but it is not at all clear that this is actually the case. This should be clarified.

Finally, I have a comment on the title and the last paragraph. These fossils are not newly discovered. Mustoe and Eberl write that locals found the Las Crucitas site sometime before 1963 when a specimen was put on display, and Eberl visited the site with the help of a local who remembered the locality. They also write that the Salitre locality was shown to Eberl by a teacher from Agua Blanca, though no indication is provided for how long the teacher has known of the site or how they learned of it.

Author Response

Thanks for the constructive suggestions for improving the manuscript. We have added additional text to address the issues that were raised in the review. These include information about the curation of specimens, the possible relationship of the fossil wood localities to material from archaeological sites, the taxonomic identification of fossil wood, and the geology of the site. The manuscript lacked clarity on these issues, so we appreciate your observations. The focus of our preliminary study was to learn whether fossil wood localities occur in Guatemala, and whether materials from sites could be distinguished based on physical characteristics. We have added some words of clarification on this point. Field work in 2020 will continue the search for addition localities, and new participants will include a paleobotanist who specializes in fossil wood, and who will hopefully be able to provide taxonomic identifications. We are also hoping to obtain more detailed information about the geologic setting, but this is challenging given the scarcity of geologic mapping. Much of the geologic research in Guatemala has focused on two subjects: mineral deposits (because of their economic value), and modern volcanoes (because of their eruptive risk to humans). Neogene paleontology hasn't received much attention, which make our research novel, but without a wealth of background geologic evidence.

George Mustoe